# Antibiotic Prevalence Study and Factors Influencing Prescription of WHO Watch Category Antibiotic Ceftriaxone in a Tertiary Care Private Not for Profit Hospital in Uganda

**DOI:** 10.3390/antibiotics10101167

**Published:** 2021-09-26

**Authors:** Mark Kizito, Rejani Lalitha, Henry Kajumbula, Ronald Ssenyonga, David Muyanja, Pauline Byakika-Kibwika

**Affiliations:** 1Department of Medicine, College of Health Sciences, Makerere University, Kampala P.O. Box 7072, Uganda; rejani.lalitha@mak.ac.ug (R.L.); pbyakika@chs.mak.ac.ug (P.B.-K.); 2Department of Microbiology, College of Health Sciences, Makerere University, Kampala P.O. Box 7072, Uganda; hkajumbula@chs.mak.ac.ug; 3Department of Epidemiology and Biostatistics, College of Health Sciences, Makerere University, Kampala P.O. Box 7072, Uganda; rssenyonga@musph.ac.ug; 4Department of Medicine, Mengo Hospital, Kampala P.O. Box 7161, Uganda; david.muyanja@mengohospital.org

**Keywords:** ceftriaxone prescription, prevalence, antibiotic stewardship, Uganda

## Abstract

**Background:** Excessive use of ceftriaxone contributes to the emergence and spread of antimicrobial resistance (AMR). In low and middle-income countries, antibiotics are overused but data on consumption are scarcely available. We aimed to determine the prevalence and factors influencing ceftriaxone prescription in a tertiary care private not-for-profit hospital in Uganda. **Methods:** A cross-sectional study was carried out from October 2019 through May 2020 at Mengo Hospital in Uganda. Patients admitted to the medical ward and who had been prescribed antibiotics were enrolled. Sociodemographic and clinical data were recorded in a structured questionnaire. Bivariate and adjusted logistic regression analyses were performed to determine factors associated with ceftriaxone prescription. **Results:** Study participants were mostly female (54.7%). The mean age was 56.2 years (SD: 21.42). The majority (187, 73.3%) presented with fever. Out of the 255 participants included in this study, 129 (50.6%) participants were prescribed ceftriaxone. Sixty-five (25.5%) and forty-one (16.0%) participants had a prescription of levofloxacin and metronidazole, respectively. Seven participants (2.7%) had a prescription of meropenem. Out of 129 ceftriaxone prescriptions, 31 (24.0%) were in combination with other antibiotics. Overall, broad-spectrum antibiotic prescriptions accounted for 216 (84.7%) of all prescriptions. Ceftriaxone was commonly prescribed for pneumonia (40/129, 31%) and sepsis (38/129, 29.5%). Dysuria [OR = 0.233, 95% CI (0.07–0.77), *p* = 0.017] and prophylactic indication [OR = 7.171, 95% CI (1.36–37.83), *p* = 0.020] were significantly associated with ceftriaxone prescription. **Conclusions:** Overall, we observed a high prevalence of prescriptions of ceftriaxone at the medical ward of Mengo Hospital. We recommend an antibiotic stewardship program (ASP) to monitor antibiotic prescription and sensitivity patterns in a bid to curb AMR.

## 1. Background

Ceftriaxone is a broad-spectrum, third-generation cephalosporin antibiotic [1]. It is used in the treatment of many community and hospital-acquired infections mainly due to *Streptococcus pneumoniae*, *Streptococcus pyogenes*, *Staphylococcus aureus*, *Escherichia coli*, *Neisseria meningitides*, *Neisseria gonorrhoeae*, *Proteus mirabilis*, and *Enterobacter* spp. [1]. Indications for ceftriaxone use include acute bacterial meningitis, severe pneumonia, complicated intra-abdominal infections, pyelonephritis, severe prostatitis, and sepsis in neonates and children [2].

Ceftriaxone use is high in many low-income countries owing to its broad-spectrum activity, low toxicity, and cheaper cost compared with other antibiotics [3,4,5]. In Uganda, studies conducted in public healthcare facilities have reported high ceftriaxone prescription rates among hospitalized patients [5,6,7].

Unfortunately, there are reports of increasing resistance to this commonly prescribed antibiotic, and this is one of the emerging global health issues. For instance, in one review on antimicrobial resistance (AMR) covering Eastern African countries, Gram-negative and Gram-positive resistance to ceftriaxone was very high [8]. In Uganda, a summary of antimicrobial drug resistance patterns from blood cultures collected between June 2013 and October 2014 at Mulago National Referral Hospital showed high resistance of *Enterobacteriaceae* to ceftriaxone [9]. Furthermore, a study done at Uganda Cancer Institute among febrile cancer patients observed that Gram-negative resistance to ceftriaxone was very high [10].

To curb the increasing AMR, the World Health Organization (WHO) recommends continuous monitoring of antibiotic use [11]. Consequently, cephalosporins are frequently identified as a particular target for use evaluation and antibiotic stewardship [12].

In Uganda, similar to many other low-income countries, the private sector is increasingly becoming an important source of health care filling gaps where no or little public health care is available [13]. Half of all Ugandans (49%) utilize the private or private-not-for-profit (PNFP) sector care [14]. Despite this, there is a paucity of data on antibiotic use in private hospitals in Uganda. This study aimed at identifying the prevalence and the factors influencing ceftriaxone prescription at a tertiary care private not-for-profit hospital in Kampala, Uganda.

## 2. Methods

### 2.1. Study Setting and Design

A cross-sectional study was carried out from October 2019 through May 2020 at the medical ward of Mengo Hospital in Uganda. Mengo Hospital is a tertiary care private not-for-profit faith-based hospital in Kampala, the capital city of Uganda, with a bed capacity of 300. The medical ward consists of 40 beds and admits an average of 8 patients in 24 h and 240 patients over a month.

### 2.2. Data Collection

A team of doctors conducted the study at the medical ward of Mengo Hospital. Training and piloting of the study instruments were conducted for the study team before the start of the study. The study was conducted on weekdays. Hospital files for patients admitted to the medical ward of Mengo Hospital were reviewed. Patients with an antibiotic prescription were approached for written informed consent to participate in the study. For participants who could not consent, their next of kin was approached. We administered a structured pretested questionnaire to the study participants or their next of kin to obtain information about their sociodemographic data (age, sex, address), and medical history. Physical examination was performed on all participants. The name of the antibiotic(s) prescribed and the indication(s) for the prescription at the time of admission were extracted from the files.

### 2.3. Data Handling and Statistical Analysis

We coded and double entered all variables into Epidata version 3.0, and thereafter exported it to STATA^®^ version 14, where the data were cleaned and analyzed. The prevalence of ceftriaxone prescription was defined as a percentage of patients with ceftriaxone prescription among the total number of patients on any systemic antibiotic. Bivariate and adjusted logistic regression analyses were performed to determine factors associated with ceftriaxone prescription. Statistical significance was set at cutoff points of 0.2 and 0.05 for bivariate and adjusted analyses, respectively.

## 3. Results

Two hundred and seventy-three patients admitted to the medical ward of Mengo Hospital between October 2019 and May 2020 received a prescription of antibiotics. Eighteen patients were excluded; fifteen did not consent while three were below 18 years of age. Out of the 255 enrolled, 129 (50.6%) were prescribed ceftriaxone as shown in Figure 1 below.

### 3.1. Demographic Characteristics of Study Participants

Among the 255 participants, 54.5% were female. The mean age was 56.2 years (SD: 21.42) with the majority (148, 58%) over 50 years of age. More than half (156, 61.2%) of study participants were Baganda by tribe.

### 3.2. Clinical Characteristics of Study Participants

The majority of study participants had a fever (see Table 1). The median temperature was 36.5 °C (IQR 36.1–37.0). The median systolic blood pressure was 125 mmHg (IQR 104–147). One hundred and eighteen participants (46.3%) had at least one co-morbidity. Diabetes mellitus was the most prevalent co-morbidity. Nearly a third had crackles on physical examination. More than half of study participants had received antibiotics 5 days before admission (see Table 1).

#### Prevalence of Ceftriaxone Use

Out of 255 study participants, 129 [(50.6%), 95% CI: 44.4–56.7%] were prescribed ceftriaxone. Of these, 40 (31.0%) were prescribed for pneumonia, 38 (29.5%) for sepsis, 14 (10.9%) for gastroenteritis, and 14 (10.9%) for urinary tract infection.

### 3.3. Prescribed Antibiotics and Indications

The most commonly prescribed antibiotics were ceftriaxone and Levofloxacin (see Table 2). Seven patients received meropenem. Of the 255 participants, 71 (27.8%) received a combination of 2 antibiotics. Ceftriaxone and metronidazole, 20 (28.2%), ceftriaxone and azithromycin, 9 (12.7%), and levofloxacin and azithromycin, 6 (8.5%) were the commonest combinations. Two participants received a combination of ceftriaxone, metronidazole, and levofloxacin. Overall, broad-spectrum antibiotic prescriptions accounted for 216 (84.7%) of all prescriptions. Pneumonia and sepsis were the most common indications for antibiotic use (see Table 2).

### 3.4. Indications for Commonly Prescribed Antibiotics

Ceftriaxone was commonly prescribed for pneumonia (40/129, 31%) and sepsis (38/129, 29.5%). Levofloxacin was commonly prescribed for sepsis (19/65, 29.2%) and UTI (17/65, 26.2%). Metronidazole was prescribed commonly for gastroenteritis (17/41, 41.5%).

### 3.5. Factors Associated with Ceftriaxone Prescription

Bivariate and multivariate analyses were performed to identify factors associated with ceftriaxone prescription. Significant factors at bivariate analysis were: sex, dysuria, skin rash, UTI indication, medical prophylactic indication, and HIV disease. Factors significantly associated with ceftriaxone prescription included dysuria and medical prophylactic indication (see Table 3).

## 4. Discussion

This study was carried out to investigate the prevalence of antibiotic prescription among medical inpatients in a tertiary care private not-for-profit hospital in Kampala, Uganda. The most commonly prescribed antibiotics were ceftriaxone (50.6%), levofloxacin (25.5%), metronidazole (16.1%), azithromycin (10.6%), and piperacillin-tazobactam (6.3%). This is comparable to what has been reported from studies done in public healthcare facilities in Uganda. For instance, a study done at Mulago national referral hospital reported prescription rates of 16% and 6% for metronidazole and azithromycin, respectively [5]. Similar results have also been reported in Tanzania [15], Ethiopia [16], and Egypt [17]. Studies investigating antibiotic prescription patterns in public healthcare facilities in Uganda have reported prescription rates for ciprofloxacin at 10.2% [18] and 19% [5]. A higher levofloxacin prescription rate in this study could likely be attributed to a change in antimicrobial susceptibility patterns. Ceftriaxone, metronidazole, azithromycin, and piperacillin-tazobactam are all listed on the 21st WHO model list of essential medicines 2019 [19] and the ministry of health essential medicines list for Uganda 2016 [20], tools designed to facilitate the appropriate use of antibiotics. Thus, the antibiotic prescription culture at Mengo Hospital is fairly adherent to international and local recommendations. Importantly, levofloxacin, the second most commonly prescribed antibiotic in this study is not listed on these lists. This highlights the need for an antibiotic prescription guideline at this facility in accordance with national and international guidelines to reduce irrational use of levofloxacin, a WHO Watch category antibiotic.

In this study, the indications for antibiotic use were pneumonia (29%), sepsis (28.2%), urinary tract infection (UTI) (14.5%), gastroenteritis (13%), and central nervous system (CNS) infection (2%). Pneumonia was the most common indication for antibiotic use in this study probably because a significant proportion (46.7%) of our study participants were over 60 years and therefore more predisposed to pneumonia [21]. Urinary tract infections are also common in this age group [22]. Our findings are in agreement with what has been reported in other studies [4,16,23,24]. A report by the Uganda National Academy of Sciences on antibiotic resistance in Uganda showed that pneumonia, septicemia, acute diarrhea, and urinary tract infections were the leading bacterial infections among hospitalized patients in Uganda [24]. Another retrospective study investigating the trends of admissions and case fatality rates among medical in-patients at a tertiary hospital in Uganda showed that pneumonia, sepsis, and gastroenteritis were among the leading causes of admissions to the infectious disease wards for a period between January 2011 and December 2014 [23]. Our findings are consistent with findings from studies done in Tanzania [15], Ethiopia [2,16], Eriteria [25], the Netherlands [26], and the US [27,28].

One in every two patients admitted to the medical ward of Mengo Hospital were prescribed ceftriaxone. This is a high prescription rate of ceftriaxone given that it belongs to the watch WHO group of antibiotics that have a high resistance potential and should not be prescribed routinely [29].Ceftriaxone is among the most commonly utilized antibiotics owing to its high potency, a wide spectrum of activity, and a low risk of toxicity [2]. It is used to treat different types of bacterial infections including pneumonia, abdominal, skin and soft tissue, and urinary tract infections [30]. It has the advantage of wide coverage of pathogens, easy administration as it is once-daily dosing–limiting the nursing time needed and a low cost compared with many other antibiotics [31,32]. This probably explains its high prescription rate in this study.

Not surprisingly, studies done in other public healthcare facilities in Uganda have shown a relatively high extent of ceftriaxone prescription; for instance, in a study from a tertiary care hospital in Kampala, the ceftriaxone prescription rate was 43% [7] and 66% [5]. Similar studies done in public healthcare facilities in other parts of Uganda such as Mbarara [33], and Bwindi [6] have reported relatively high ceftriaxone prescription rates of 77.7% and 45%, respectively. Similarly, studies were done in private healthcare facilities in Ethiopia—49% [34], Pakistan—46% [35], and Bangladesh—50% [36]; countries with social and economic indicators of development similar to those of Uganda [37] have also reported a relatively high ceftriaxone prescription rate.

Thus, generally, the observed ceftriaxone use in the present study is irrationally high as this drug should only be used in confirmed severe infections. Overuse of ceftriaxone contributes to antibiotic resistance [38]. To reduce ceftriaxone use and mitigate the rapid development of AMR, hospital antimicrobial stewardship programs should impose and enforce prescription restrictions, set up antibiotic consumption surveillance systems, and deliver appropriate education campaigns to prescribers. Stewardship will increase knowledge on resistance patterns to secure other low-spectrum antibiotics that can still be effectively used. Notably, there is no available published data on resistance rates of bacteria to ceftriaxone at Mengo Hospital. However, studies done in other public healthcare facilities in Uganda have reported resistance rates of 64–66% for *Escherichia coli* [9,10] and 85–100% for Klebsiella pneumoniae [9,10]. It is likely that the resistance rates of bacteria to ceftriaxone at Mengo Hospital are similarly high.

We further investigated factors that may influence ceftriaxone prescription. Factors that were significantly associated with ceftriaxone use included a history of pain on passing urine (dysuria) and medical prophylactic indication. Attending clinicians were 76% less likely to prescribe ceftriaxone to patients with dysuria compared with those with no dysuria. It is not surprising that participants with dysuria were less likely to be prescribed ceftriaxone by the attending clinicians. Dysuria is a well-recognized symptom of urinary tract infections [39]. International [40] and local [41] guidelines recommend nitrofurantoin monohydrate and fluoroquinolones as the first line in the treatment of both uncomplicated and complicated urinary tract infections (acute pyelonephritis), respectively, in adults, and therefore it is likely that clinicians followed these guidelines. Clinicians were seven times more likely to prescribe ceftriaxone for medical prophylaxis compared with sepsis. In our study, medical prophylaxis referred to conditions where antibiotics were prescribed without clinical and/or laboratory evidence of infection for example ceftriaxone prescribed for patients with liver cirrhosis complicated by ascites without evidence of infection to prevent development of spontaneous bacterial peritonitis or antibiotics prescribed for patients with diabetes mellitus and hyperglycemia but no evidence of infection. This implies that whenever clinicians assess patients and find no evidence of infection, they are likely to prescribe ceftriaxone for prophylaxis. This is probably because it is a broad-spectrum antibiotic with activity against a wide range of pathogens. Moreover, qualitative studies done in other healthcare facilities revealed that sometimes clinicians prescribe antibiotics for fear of bad outcomes [42,43,44]. It is therefore likely that ceftriaxone prescribed for medical prophylaxis was unnecessary. Notably, prophylactic use of antibiotics is a potential driver of antimicrobial resistance [45,46,47,48] which is associated with adverse outcomes. This highlights the importance of having a local prescription guideline and whenever appropriate the need to wait for laboratory results before choosing the best antimicrobial therapy.

A strength of this study is that data were collected from a private tertiary care facility, and it could be one of the first studies investigating ceftriaxone prescription in such a setting. This could imply and depict the real practice in many private tertiary hospitals in Uganda. Therefore, our paper presents data from an area and setting of the world with limited available data. We believe it adds valuable information to antibiotic prescribing and stewardship in private healthcare facilities in low-income countries. Results from this study were presented to clinicians and stakeholders at Mengo Hospital and an antimicrobial stewardship committee was set up to promote appropriate antibiotic use at Mengo Hospital.

However, we acknowledge the inherent weaknesses of this study in that first, the prescriptions were strictly based on the physicians’ clinical acumen. Second, there are other factors we could not investigate such as influence from pharmaceutical and health insurance companies. It would be important to explore the contribution of these factors toward excessive use of antibiotics in private tertiary healthcare facilities in Uganda. Additionally, this was a single site study and therefore more data may need to be collected from other sites before generalizing the results.

## 5. Conclusions

Prevalence of ceftriaxone prescription was high in this hospital, mainly for pneumonia and sepsis. We recommend an ASP to monitor antibiotic prescription and sensitivity patterns in a bid to curb AMR.

## Figures and Tables

**Figure 1 antibiotics-10-01167-f001:**
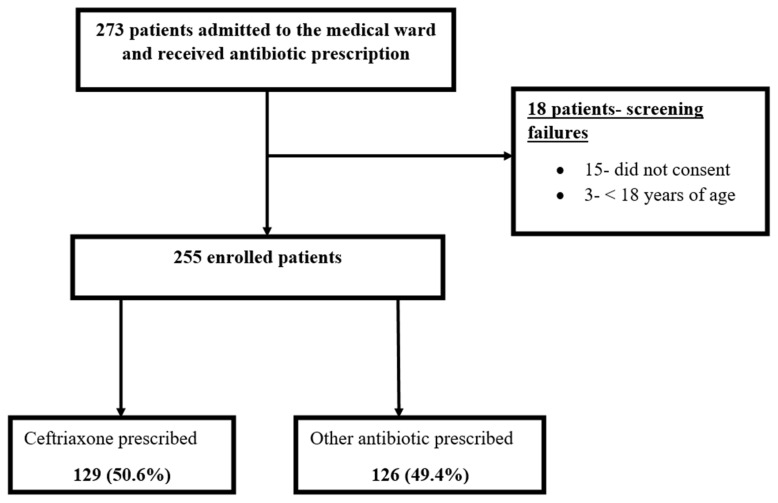
Study profile of hospitalized patients with antibiotic prescription.

**Table 1 antibiotics-10-01167-t001:** Clinical characteristics of the study participants.

	Frequency (n)	Percentage (%)
**Symptoms**		
History of fever	187	73.3
History of joint pains	107	42.0
History of cough	103	40.4
History of muscle pains	91	35.7
History of vomiting	90	35.3
History of abdominal pain	86	33.7
History of rapid breathing	78	30.6
History of confusion	77	30.2
History of passing urine frequently	37	14.5
History of wound or ulcer	27	10.6
History of pain on passing urine	20	7.8
History of convulsions	19	7.5
**Co-morbidities**		
Diabetes mellitus	53	20.8
HIV	39	15.3
Stroke	10	3.9
Chronic lung disease	9	3.5
Malignancy	8	3.1
Heart disease	7	2.7
Chronic liver disease	5	2.0
Chronic kidney disease	3	1.2
**Use of antibiotics 5 days prior to admission**	145	56.9
**Clinical Signs**		
Pallor ^$^	101	39.6
Lymphadenopathy	5	2.0
Oral thrush	8	3.1
Skin rash	10	3.9
Wound or ulcer	25	9.8
Bronchial breathing	9	3.5
Crackles	73	28.6
Rhonchi	15	5.9
Temperature ≥ 37.5 °C	54	21.2
Pulse > 100 beats/min	116	45.5
Systolic blood pressure ≤ 100 mmHg	51	20.0
Respiratory rate ≥ 22 breaths/min	70	27.7
**Glasgow Coma Scale (GCS)**		
>13	204	80.0
≤13	51	20.0
**Quick Sequential Organ Failure Assessment (q SOFA) score**		
0	127	49.8
1	92	36.1
2–3	36	14.1

^$^ Pallor (Mild/Moderate 81, Severe 20).

**Table 2 antibiotics-10-01167-t002:** Prescribed antibiotics and indications for antibiotic use.

	Frequency (N = 255) (n)	Percentage (%)
**Antibiotics**		
Ceftriaxone	129	50.6
Levofloxacin	65	25.5
Metronidazole	41	16.1
Azithromycin	27	10.6
Piperacillin-tazobactam	16	6.3
**Amoxicillin/clavulanic acid**	16	6.3
Gentamycin	6	2.4
Others *	28	11.0
**Indication**		
Pneumonia	74	29.0
Sepsis	72	28.3
UTI	37	14.5
Gastroenteritis	33	12.9
Medical prophylaxis **	16	6.3
Others ***	23	9.0

* Cefuroxime (5), Meropenem (5), Flucamox (4) Septrin (3), Ciprofloxacin (2), Imipenem (2) Salbactam (2), Ampiclox (1), Cefazolin (1), Cefixime (1), Cefpodoxime (1) Erythromycin (1). ** Refers to conditions where antibiotics were prescribed without clinical and/or laboratory evidence of infection for example liver cirrhosis complicated by ascites to prevent development of spontaneous bacterial peritonitis. *** CNS infection (5), Cellulitis (5), Sinusitis (2), Tonsillitis (2), PID (1), Abscess (1), Cholecystitis (1), Conjunctivitis (1), Empyema (1), Oral Abscess (1), Pharyngitis (1), Rhino sinusitis (1), and Septic arthritis (1).

**Table 3 antibiotics-10-01167-t003:** Multivariate logistic analysis for factors associated with ceftriaxone prescription.

	Adjusted Odds Ratio	(95% Confidence Interval)	*p*-Value
**Sex**			
Male	1		
Female	0.668	(0.39–1.14)	0.145
**History of pain on passing urine**			
No	1		
Yes	0.233	(0.07–0.77)	**0.017** *
**Indication**			
Sepsis	1		
Pneumonia	1.079	(0.55–2.11)	0.825
UTI	0.989	(0.40–2.44)	0.982
GIT Infection	0.670	(0.29–1.61)	0.391
Medical prophylaxis	7.171	(1.36–37.83)	**0.020** *
Others	0.588	(0.22–1.59)	0.296
**HIV**			
No	1		
Yes	1.792	(0.81–3.97)	0.150
**Skin Rash**			
No	1		
Yes	4.671	(0.53–41.22)	0.165

* Statistically significant at 5% level of significance.

## Data Availability

The datasets used and/or analyzed during the current study are available from the corresponding author on reasonable request.

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
