# Peer review of "Antibiotic Prevalence Study and Factors Influencing Prescription of WHO Watch Category Antibiotic Ceftriaxone in a Tertiary Care Private Not for Profit Hospital in Uganda"

_antibiotics, 2021, doi:10.3390/antibiotics10101167_

Round 1

Reviewer 1 Report

Thank you for the opportunity to review this paper. The topic is very interesting and very necessary due to concerns about antimicrobial management. However, I believe that more data need to be collected and the cause-and-effect relationship between symptoms and ceftriaxone prescribing should should be explained more clearly in order to increase the significance of this paper.

Line 136-137 -  "The majority (187, 73.3%) of study participants had a fever. The median temperature was 136 36.5oC (IQR 36.1-37.0)." I don't understand these 2 sencentces - I don't understand these two sentences - since when is temperature 36.5C  considered as a fewer? Or did you want to say something else, but it remained so vaguely written?

References  

The authors did not follow the instructions of the journal for citing references. Therefore, I suggest them to correct the stlye of referencing, beceause now they are not in the appropriate format. Also, references are not written uniformly, some of them have abbreviated journal names and some of them full names.

Limitations 

Your study has many limitations, but I can't see them written in the text. Limitation section is missing, so please provide it. Also, you should more clearly point it out what is actually so new and significant in the article that it should be interesting to the future readers? Therefore, I suggest authors to think about it and add it to the manuscript. 

Author Response

Response to Reviewer 1 comments

Point 1: Thank you for the opportunity to review this paper. The topic is very interesting and very necessary due to concerns about antimicrobial management. However, I believe that more data need to be collected and the cause-and-effect relationship between symptoms and ceftriaxone prescribing should be explained more clearly in order to increase the significance of this paper.

Response 1: Thank you for this suggestion. It would be useful to collect more data. However, this paper analyzes data that was collected as part of my thesis’ work. The cause- and-effect relationship between symptoms and ceftriaxone prescribing was analyzed at bivariate level and, dysuria, and skin rash were the significant factors. Factors that were significantly associated with ceftriaxone use included a history of pain on passing urine (dysuria) and prophylactic indication. Attending clinicians were 76% less likely to prescribe ceftriaxone to patients with dysuria compared to those with no dysuria- page 12 lines 245-246.

Point 2: Line 136-137 -  "The majority (187, 73.3%) of study participants had a fever. The median temperature was 136 36.5oC (IQR 36.1-37.0)." I don't understand these 2 sentences - I don't understand these two sentences - since when is temperature 36.5C  considered as a fever? Or did you want to say something else, but it remained so vaguely written?

Response 2: Thank you for pointing this out. These two sentences are separate from one another presenting results that were obtained. I agree, temperature of 36.5C is not considered as fever.

Point 3: The authors did not follow the instructions of the journal for citing references. Therefore, I suggest them to correct the style of referencing, because now they are not in the appropriate format. Also, references are not written uniformly, some of them have abbreviated journal names and some of them full names.

Response 3: We agree with this and have accordingly corrected the style of referencing. The references are now written uniformly as per the referencing style recommended by the journal- pages 16- 18.

Point 4: Your study has many limitations, but I can't see them written in the text. Limitation section is missing, so please provide it. Also, you should more clearly point it out what is actually so new and significant in the article that it should be interesting to the future readers? Therefore, I suggest authors to think about it and add it to the manuscript. 

Response 4: Thank you for this suggestion. The study limitations are outlined on page 13 lines 259- 263. “However, we acknowledge the inherent weaknesses of this study in that first, the prescriptions were strictly based on the physicians’ clinical acumen. Second, there are other factors we could not investigate such as influence from pharmaceutical and health insurance companies. It would be important to explore the contribution of these factors toward excessive use of antibiotics in private tertiary healthcare facilities in Uganda”

This could be one of the first studies investigating antibiotic prescription in private healthcare facilities in Uganda. Therefore, our paper presents data from an area and setting of the world with limited available data. We believe it adds valuable information to antibiotic prescribing and stewardship in private healthcare facilities in low-income countries.

Reviewer 2 Report

This report analyzes usage data in an area and setting of the world with limited available data - because of this - the report adds valuable information. 

Introduction: 

  • (page 3, lines 57 to 68) the introduction is generally well written but includes excessive information for an introduction. Please remove most of the specific percentages and other numbers listed and instead list general statement suitable for an introduction. (example: In many eastern African countries, Gram-positive and Gram-negative resistance to ceftriaxone is very high (references)).

Methods:

  • Not sure why religion was collected; doesn't seem like it adds to the paper at all unless there is some reason why religious affiliation is associated with antibiotic use
  • Throughout the results, please remove duplicated data (data typed into the text AND also listed in the exact same form in a table; you can refer to the table without listing the actual data; (example: A majority of patients had fever (see table 1).)
  • its unclear if the antibiotics listed were prescribed just at admission, were continued for a certain duration, or were all antibiotic prescribed during that hospitalization. this should be detailed in the methods. 
  • If all antibiotics during the admission were collected, duration of therapy should be listed as well (average duration and percentage of hospital stay with antibiotics: days per 1000 patient days). 
  • Please describe what is meant by 'prophylactic' indication; is this surgical prophylaxis or something else? 
  • Line 71 "statistical significance at the bivariate..." is redundant with the next sentence - please remove. again, please do not duplicate the values in teh text if listed in the table
  • While the study has not aimed to determine appropriateness of use - it would be useful to see some information: number of patients with positive bacterial cultuures; number of patients that had therapy de-escalated to something narrower, etc

Discussion

  • I suggest included additional detailed information in the discussion about specific resistance profile at the study hospital; what is the rate of Ecoli resistance to ceftriaxone; Streptococcus pneumoniae?
  • reference #39 is not an appropriate reference. Please include a link to international guidelines on treatment of uncomplicated cystitis (WHO, IDSA, or European counterpart are considered international generally); these guidelines do NOT recommend fluoroquinolones as first line therapy for uncomplicated cystitis. 
  • citations likely need to be reformated - some references may not be appropriate (eg. there are higher quality references that could be used)

Author Response

Response to Reviewer  2  Comments

Introduction

Point 1: (Page 3, lines 57 to 68) the introduction is generally well written but includes excessive information for an introduction. Please remove most of the specific percentages and other numbers listed and instead list general statement suitable for an introduction. (example: In many eastern African countries, Gram-positive and Gram-negative resistance to ceftriaxone is very high (references)).

Response 1: Thank you for pointing this out. We agree with this comment. Therefore we have deleted the percentages and numbers as suggested- Page 3 lines 57 to 68.

Methods and results

Point 2: Not sure why religion was collected; doesn't seem like it adds to the paper at all unless there is some reason why religious affiliation is associated with antibiotic use

Response 2: Thank you for pointing this out. Religion doesn’t seem to add more information to the paper. Information on religion has been removed- page 5 line 94, page 6 lines 132-134.

Point 3: Throughout the results, please remove duplicated data (data typed into the text AND also listed in the exact same form in a table; you can refer to the table without listing the actual data; (example: A majority of patients had fever (see table 1).)

Response 3:  We agree with this and have accordingly incorporated your suggestion throughout the results section of the manuscript- pages 6-10.

Point 4:  it’s unclear if the antibiotics listed were prescribed just at admission, were continued for a certain duration, or were all antibiotic prescribed during that hospitalization. this should be detailed in the methods. 

Response 4: Agree. We have accordingly revised the methods to emphasize this point. The antibiotics listed were prescribed at admission- page 5 line 96. We did not follow up the patients to obtain information on duration of hospital stay and change of antibiotics.

Point 5: If all antibiotics during the admission were collected, duration of therapy should be listed as well (average duration and percentage of hospital stay with antibiotics: days per 1000 patient days).

Response 5: Thank you for this suggestion. It would have yielded more useful information. However, data on duration of therapy and all antibiotics prescribed during hospital stay were not collected as it was not part of the study.

Point 6: Please describe what is meant by 'prophylactic' indication; is this surgical prophylaxis or something else? 

Response 6: Thank you for pointing this out. Prophylactic indication in our study meant conditions where antibiotics were prescribed without clinical or laboratory evidence of infection.( Example ceftriaxone was prescribed for patients with liver cirrhosis without evidence of infection to prevent against development of Spontaneous bacterial peritonitis, antibiotics were prescribed in patients with diabetes mellitus and hyperglycemia).

Point 7: Line 71 "statistical significance at the bivariate..." is redundant with the next sentence - please remove. again, please do not duplicate the values in teh text if listed in the table

Response 7: Agree, we have accordingly removed this statement- page 9, line 172.

Point 8: While the study has not aimed to determine appropriateness of use - it would be useful to see some information: number of patients with positive bacterial cultures; number of patients that had therapy de-escalated to something narrower, etc

Response 8: Thank you for this suggestion. It would have been interesting to explore this aspect. However, in the case of our study, it seems slightly out of scope because the study was not aimed at determining appropriateness of antibiotic use. We did not collect data on therapy de-escalation and number of patients with positive bacterial cultures.

Discussion

Point 9: I suggest included additional detailed information in the discussion about specific resistance profile at the study hospital; what is the rate of E.coli resistance to ceftriaxone; Streptococcus pneumoniae?

Response 9: Thank you for this suggestion. It would be useful to include information in the discussion about the specific resistance profiles at Mengo Hospital. However, there is no published literature on resistance profiles at Mengo Hospital. Published studies report data from public healthcare facilities such as Mulago Hospital. In these studies, Escherichia coli resistance rates to ceftriaxone range from 64- 66% while Klebsiella pneumoniae resistance to ceftriaxone rates range from 85-100%. Most studies reporting Streptococcus pneumoniae resistance to ceftriaxone rates were done among pediatric populations and report no resistance to ceftriaxone. It is likely that the resistance rates of bacteria to ceftriaxone at Mengo Hospital are similar- page 11 lines 222- 227.

Point 10: reference #39 is not an appropriate reference. Please include a link to international guidelines on treatment of uncomplicated cystitis (WHO, IDSA, or European counterpart are considered international generally); these guidelines do NOT recommend fluoroquinolones as first line therapy for uncomplicated cystitis.

Response 10: Thank you for pointing this out. IDSA guidelines recommend nitrofurantoin monohydrate as the first line therapy for uncomplicated cystitis and fluoroquinolones as first line therapy for acute pyelonephritis. Reference #39 has accordingly been replaced with a higher quality reference- page 12 lines 242-245 and page 16 line 438.

Point 11: citations likely need to be reformated - some references may not be appropriate (eg. there are higher quality references that could be used)

Response 11: We agree with this and have accordingly reformatted the citations- page 16. We have also included higher quality references- page 16 line 438.

Round 2

Reviewer 1 Report

As the authors have accepted all suggestions from reviewer and made corrections (those that they could correct beside of the limitations of the study), I have no further objections to the text of the manuscript. The manuscript has been much improved and it is now easy to read. 

Reviewer 2 Report

While the study has methodological limitations that persist, the authors have adequately responded to all reviewer comments.